

**PeerJ Hubs**
Published on behalf of

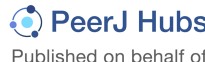
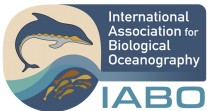

# Diet and prey selectivity in co-occurring eelpout fish and bythograeid crabs in a deep-sea hydrothermal vent community

Deidric B. Davis[1] and Nancy Smith[2]

[1] Duke University Marine Lab, Beaufort, NC, United States of America
[2] Marine Science, Eckerd College, St. Petersburg, FL, United States of America

## ABSTRACT

Understanding the trophic ecology of deep-sea communities is central to assessing ecological structure and function, which is often lacking in remote oceanographic environments such as hydrothermal vents. Using stomach content analysis coupled with published stable isotope data, we assessed diet and prey selectivity in two common predators, eelpouts (*Pyrolycus manusanus*) and crabs (*Austinograea alayseae*), from a South Pacific deep-sea hydrothermal vent community. Using specimens collected during a cruise in 2007, we found that eelpouts strongly preferred alvinocarididshrimp. This observation is s upported by the Ivlev index, which measures the selection of prey in relation to their abundance or availability. Crabs exhibited a diverse diet, including polychaetes and shrimp, suggesting a scavenging or omnivorous feeding strategy. Due to the lack of intact stomach contents in the crab, we were unable to apply the Ivlev method to quantify its prey selectivity. Our results emphasize the need to combine stomach contents, stable isotope analysis, and other complementary methodologies, to elucidate the role of predators in deep-sea food webs. In sum, our study underscores the importance of direct stomach content examination in revealing trophic relationships in hydrothermal vent systems.

## INTRODUCTION

Understanding the feeding ecology of marine animals is essential for elucidating their roles in trophic interactions, energy flow, and community structure within ecosystems (*Hayden et al., 2019*). Prey selectivity, a key aspect of feeding ecology, reveals how predators influence prey populations and the cascading effects on ecological stability and biodiversity (*Li, Wetterer & Hairston Jr, 1985*; *Schmitz, Beckerman & O'Brien, 1997*; *Cupples et al., 2011*; *Chen et al., 2021*). By selecting prey based on size, nutritional value, or availability, predators regulate energy transfer and shape the composition of their communities (*Gerking, 2014*; *Chen et al., 2021*; *Vinterstare et al., 2023*).

In the deep sea, prey selectivity is particularly significant due to the scarcity of biological resources, driving the evolution of specialized feeding strategies. For example, deep-sea predators such as zoarcid fish exhibit preferences that highlight their roles as apex

Corresponding author
Deidric B. Davis,
dbdavis@eckerd.edu,
dbd22@duke.edu

predators, which can redefine previously held assumptions about food web structures in these systems (*Heger & Sutton, 2008*; *Sancho et al., 2005*). Understanding prey selectivity not only elucidates trophic interactions, but also provides critical insights into ecosystem resilience and responses to environmental disturbances, underscoring its importance in conservation efforts (*Ripple et al., 2014*).

Stomach content analysis (SCA) is a primary tool in trophic ecology, revealing dietary composition and prey selection (*Baker, Buckland & Sheaves, 2014*). SCA provides valuable insights into the recent diet of marine organisms by identifying specific prey items, offering a direct and detailed assessment of dietary composition. This method allows for the precise identification and quantification of ingested prey, facilitating a comprehensive understanding of an organism's feeding habits and prey preferences (*Hyslop, 1980*). Additionally, SCA provides critical information on trophic interactions and food web dynamics, shedding light on the ecological roles of species within their communities (*Baker, Buckland & Sheaves, 2014*). By analyzing temporal and spatial variations in diet, SCA also helps researchers assess how environmental factors influence feeding behavior, contributing to a broader understanding of ecosystem functioning (*McMeans et al., 2019*). SCA serves as a benchmark for validating alternative dietary assessment methods, such as stable isotope and DNA-based analyses, ensuring the accuracy and reliability of dietary studies (*Carreon-Martinez & Heath, 2010*). Thus, these advantages make SCA an essential tool in understanding the feeding ecology of marine environments.

Despite these benefits, SCA has several limitations. For example, SCA only reflects an organism's diet over a short period before capture, making it unable to represent long-term dietary patterns, which can skew understanding of trophic dynamics (*Cortés, 1997*). Additionally, the digestion process often degrades or partially digests prey items, leading to potential underrepresentation of soft-bodied organisms that are more difficult to identify in stomach contents (*Hyslop, 1980*). SCA is also labor-intensive and can require large sample sizes to capture diet variability, which may be challenging in studies of rare or hard-to-collect species (*Amundsen & Sánchez-Hernández, 2019*).

Stable isotope analysis (SIA) of consumer tissues is also often used in studies of trophic ecology and is especially valuable in the study of deep-sea systems, where direct observation of feeding behaviors is often impossible (*Smith Jr & Baldwin, 1997*). SIA offers a time-averaged estimate of the trophic level of an organism ($\delta^{15}$N) and potential prey items ($\delta^{13}$C), but usually cannot identify prey to the species level. This restricts understanding of dietary interactions (*Kelly, 2000*; *Bearhop et al., 2004*). Additionally, the method is sensitive to variations in baseline isotopic values across ecosystems, which may lead to inaccuracies unless carefully accounted for (*Post, 2002*). Overlapping isotopic signatures among prey species can also obscure dietary interpretations, highlighting the need for complementary methods like SCA for a more detailed dietary profile (*Phillips et al., 2014*). Integrating multiple dietary assessment methods enhances our understanding of complex trophic relationships and ecological roles within marine systems. When SCA is combined with SIA, species' positions within the food web can be more accurately assessed, providing insights into both short-term feeding habits and long-term trophic integration (*Shin et al., 2022*).

Eelpout fish and bythograeid crabs are key components of deep-sea chemosynthetic ecosystems, though the study of their feeding ecology remains challenging due to the logistical difficulties of accessing these remote and extreme environments (*Canals et al., 2021*). Hydrothermal vent environments, with their extreme conditions of temperature, pressure, and toxicity, shape the evolutionary adaptations and feeding strategies of species residing there (*Chapman, 2018*). This study focuses on two hydrothermal vent inhabitants—an eelpout (*Pyrolycus manusanus*) and a crab (*Austinograea alayseae*)—to explore their feeding ecology and prey selectivity in a South Pacific vent community.

*Pyrolycus manusanus* (Family Zoarcidae) is a scaleless fish that is endemic to hydrothermal vents in the Southwest region of the Pacific Ocean (*Machida & Hashimoto, 2002*). As benthic predators, *P. manusanus* have a laterally compressed and elongated body (length range: 14.75 cm to 18 cm), adapted to living in crevices and tight spaces (*Machida & Hashimoto, 2002*). They have gelatinous flesh which is a common trait of deep-sea fish (*Gerringer et al., 2017*). *Pyrolycus manusanus* separates itself from other scaleless eelpouts by having an occipital pore, a distinguishing taxonomic feature (*Machida & Hashimoto, 2002*). They have a terminal mouth that is used for active prey acquisition (*Ferry, 1997*). Members of the family Zoarcidae, including *Pyrolycus manusanus*, play key roles in deep-sea chemosynthetic ecosystems, feeding on crustaceans, molluscs, and other fish while serving as the most abundant predator species (*Ferry, 1997*; *Micheli et al., 2002*; *Sancho et al., 2005*; *Frable et al., 2023*). Studies of related species, such as *Bothrocara brunneum* and and *Bothrocara zestum* in the eastern Bering Sea reveal dietary specialization despite morphological similarities (*Stevenson & Hibpshman, 2010*). *Bothrocara brunneum* primarily consumes shrimps and mysids, while *Bothrocara zestum* is predominantly piscivorous, feeding on bathylagids and other zoarcids (*Stevenson & Hibpshman, 2010*). Such insights highlight the complexity of trophic interactions in deep-sea systems, emphasizing the need for further comparative studies in chemosynthetic habitats.

*Austinograea alayseae*, a blind crab from family Bythograeidae, is a notable inhabitant of Pacific hydrothermal vents (*Guinot & Segonzac, 2018*). This species is characterized by its pale carapace, chelae with distinct spots, and carapace length ranging from 5–15 cm (*Guinot, 1989*). Distinguishing *A. alayseae* from other *Austinograea* species requires molecular analysis and detailed examination of reproductive structures, such as gonopods, which exhibit unique morphological traits (*Leignel, Hurtado & Segonzac, 2017*; *Guinot & Segonzac, 2018*). Like other bythograeids, *A. alayseae* has adapted to the extreme environments of hydrothermal vent systems, relying on specialized feeding strategies to exploit available resources.

Bythograeid crabs are primarily scavengers and opportunistic predators, feeding on a variety of organic material found around vent communities. Their diet typically includes vent-endemic species such as polychaetes, small molluscs, and other crustaceans, as well as microbial mats that proliferate near vent openings (*DeBevoise, Childress & Withers, 1990*; *Zhang et al., 2017*; *Guinot & Segonzac, 2018*). These crabs have specialized adaptations for shredding and grinding their food, enabling them to process tough or coarse organic material (*Martin, Jourharzadeh & Fitterer, 1998*). Their feeding habits are shaped by the highly variable nature of hydrothermal vent ecosystems, where food availability depends

on the proximity to vent emissions and the density of surrounding biota (*Lutz & Kennish, 1993*; *Fisher, Takai & Le Bris, 2007*).

Studies have shown that bythograeids play a crucial role in vent ecosystems by contributing to the recycling of organic material, acting as scavengers that prevent the accumulation of detritus (*Van Dover, 2000*; *Zhang et al., 2017*). Their ability to exploit a wide range of food sources ensures their survival in environments with fluctuating energy inputs, making them key players in maintaining the ecological balance of vent communities (*Leignel, Hurtado & Segonzac, 2017*; *Van Dover, 2000*). These dietary strategies, combined with their morphological adaptations, highlight the evolutionary success of *A. alayseae* and its relatives in one of Earth's most extreme habitats.

Both *P. manusanus* and *A. alayseae* cohabit the South Su hydrothermal vent within the Manus Basin, a microhabitat dominated by gastropods of the genus, *Ifremeria*, which harbor chemoautotrophic bacteria (*Collins, Kennedy & Van Dover, 2012*; *Van Audenhaege et al., 2019*). Stable isotope analysis places both species as high-trophic omnivores but does not capture fine-scale dietary preferences or prey selectivity (*Collins, Kennedy & Van Dover, 2012*). Hydrothermal vent ecosystems, such as those in the North Fiji Basin, illustrate how vent fauna partition resources and niches to sustain coexistence within small, energy-limited environments. For instance, vent crabs, as apex predators, exploit diverse prey, while other taxa specialize in specific carbon sources or microhabitats (*Suh et al., 2022*). To better understand the feeding ecology and diet of *P. manusanus* and *A. alayseae*, we conducted SCA by direct examination. For eelpouts, we employed the Ivlev Index to evaluate their dietary composition in relation to prey availability (*Bardach, 1962*).

# MATERIALS & METHODS

## Sample selection

To characterize the stomach contents of eelpout, *Pyrolycus manusanus*, and crab, *Austinogrea alayseae*, individuals were collected as by-catch in 2007 from the South Su hydrothermal vent aboard the CS *Wave Mercury*, during the Luk Luk cruise, a partnership between Duke University and Nautilus Minerals. During the cruise, invertebrate benthic samples across ~22 taxa were collected by the Perry Slingsby ROV, modified for biological sampling and equipped with a slurp-gun as described in *Collins, Kennedy, & Van Dover (2012)*. The quantitative species-abundance matrices from the *Collins, Kennedy & Van Dover (2012)* study were used as a baseline against which we assessed prey selectivity. Specimens were collected from a site known as IF2 (3.811°S, 152.104°E) at a depth of 1,399 m, and subsequently vouchered at Duke University Marine Laboratory. Specimens were fixed in 10% borax-buffered formalin and preserved with 70% ethanol. A total of 7 eelpouts (mean length, 16.21 cm ± 1.18 SD) and 25 crabs (carapace length > 2.6 cm) were used for stomach content analysis, as described below.

## Stomach content collection and analysis

*Pyrolycus manusanus* and *Austinogrea alayseae* samples were carefully dissected and the gastro-intestinal tissue was pulled apart by forceps to expose stomach contents. Preserved digested materials were observed under a microscope, and all items were photographed

 

and kept for future identification. Prey items were identified to their lowest taxonomic level, and recognizable items were counted. Non-organic materials including plastic and sulfides were discarded and were not used in any subsequent analyses. As crab stomach contents were shredded by their mandibles, this made the total number of individuals in the stomach contents indeterminable. Thus, the presence (not abundance) of prey items was noted for crabs.

## Ivlev's index

Ivlev's index (*Bardach, 1962*) of selectivity is a measure to determine the proportion of prey selection with respect to its availability. It has been used in vertebrates, especially fish (*O'Brien & Vinyard, 1974*). The index (E) is the proportion of an item in the stomach (r1) minus the proportion of the item in the environment (p1) divided by the proportion of an item in the stomach (r1) plus the proportion of the item in the environment (p1). Prey availability (p1) was derived from the species-abundance matrix of *Collins, Kennedy & Van Dover (2012)* for the *Ifremeria* habitat. The equation is described as $E = (r1-p1)/(r1+p1)$. The Ivlev index (E) has a range of $-1$ to 1. When the *E* value is closer to 1, this shows high prey selectivity, while a value of $-1$ means the prey is least likely to be selected by its predator. When *E* values are 0, prey selection is random or non-selective. After calculating *E* values, we determined if prey selectivity was influenced by eelpout size (body length) by conducting a regression analysis between eelpout body length and Ivlev values for all prey including alvinocaridid shrimp, *Lepetodrilus schrolli* (small limpet), *Shinkailepas tollmanni* (large limpet) and *Eochionelasmus ohtai* (barnacle). Eelpouts of different body lengths have been shown to ingest different prey, and thus, predator size can influence prey selectivity and stomach content composition (*Gyldenskog, 2019*).

## Taxa identification

Prey items were identified to the lowest possible taxonomic level using morphological characteristics, cross-referenced with existing taxonomic keys and the World Register of Marine Species. In cases where digestion obscured diagnostic features, identifications were inferred based on remaining morphological traits, ecological context, and habitat assessment as described by *Collins, Kennedy, & Van Dover (2012)*, who referred to *Desbruyères, Hashimoto, & Fabri (2006)* and taxonomic experts for identification. For instance, if a barnacle remnant was found in a predator's stomach contents, we referenced *Collins, Kennedy, & Van Dover (2012)* who documented only a single barnacle species in this habitat, allowing us to reasonably infer the remnant belonged to that species. When necessary, the most probable taxonomic assignment was made to account for fragmentation due to digestive processes.

## RESULTS

A total of 22 taxa (13,199 individuals) were observed at South Su by *Collins, Kennedy, & Van Dover (2012)*, and of these, four taxa (32 individual prey items) were present in the stomach contents of eelpout, *Pyrolycus manusanus,* and 10 taxa (unknown quantity) were present in the stomach contents of the crab, *Austinogrea alayseae* (Table 1). Four taxa

**Table 1  Taxa abundance observed in the stomach contents of eelpout and crab.** Number of individuals observed at South Su Manus Basin and in the gut contents of eelpout (*Pyrolycus manusanus*, $n = 7$) and crab (*Austinograea alayseae*, $n = 25$).

| Taxa present | South Su[a] | Eelpout | Brachyuran |
|---|---|---|---|
| Porifera | | | |
| *Abyssocladia sp.* | 6 | 0 | 0 |
| Cnidaria | | | |
| *Keratoisis sp.* | 1 | 0 | 1 |
| Mollusca | | | |
| *Neomphalid n. gen., n. sp.* | 166 | 0 | 0 |
| *Bathyacmaea jonassoni* | | 0 | 0 |
| *Lepetodrilus schrolli* | 11,360 | 9 | 2 |
| *Shinkailepas tollmanni* | 859 | 3 | 0 |
| *P uncturella sp.* | 1 | 0 | 0 |
| *Alviniconcha spp.* | 1 | 0 | 0 |
| *Ifremeria sp.* | 131 | 0 | 0 |
| Annelida | | | |
| *Hesionidae sp.* | 1 | 0 | 0 |
| *Branchinotogluma sp.* | 3 | 0 | 1 |
| *Branchinotogluma segonzaci* | 1 | 0 | 0 |
| *Branchinotogluma trifurcus* | 109 | 0 | 0 |
| *Thermopolynoe branchiata* | 7 | 0 | 0 |
| *Prionspio sp.* | 2 | 0 | 0 |
| *Amphisamytha cf. galapagensis* | 327 | 0 | 9 |
| Arthropoda | | | |
| *Eochinoelasmus ohtai* | 25 | 4 | 7 |
| *Chorocaris sp.* | 173 | 16 | 13 |
| *Amphipoda spp.* | 2 | 0 | 2 |
| *Alvinocaris sp.* | 4 | 0 | 0 |
| *Austinograea alayseae* | 19 | 0 | 0 |
| Totals | | | |
| Total no. of taxa | 22 | 4 | 10 |
| Total no. of individuals | 13,199 | 32 | 35 |

**Notes.**
[a] Raw data from *Collins, Kennedy & Van Dover (2012)*.

were observed in both species including, *Shinkailepas tollmanni* (large limpet), *Lepetodrilus schrolli* (small limpet), alvinocaridid shrimp, and *Eochinoelasmus ohtai* (barnacle) (Table 1, Fig. 1). Eelpout stomach contents were dominated by alvinocaridid shrimp *and Lepetodrilus schrolli,* followed by *Shinkailepas tollmanni* (Table 1). Crab stomach contents had pieces of an unidentified crustacean in the highest proportion, followed by alvinocaridid shrimp and an unidentified polychaete (Fig. 2).

Eelpouts showed the highest selectivity for alvinocaridid shrimp (Fig. 1A), as $E$ values were high, ranging from 0.9 to 1.0 (Table 2). The other three taxa (*Shinkailepas tollmanni*, *Lepetodrilus schrolli*, *Eochinoelasmus ohtai*) were selected less by eelpout (negative $E$ values; Table 2) despite their high abundance at South Su (Table 1). Observations show that

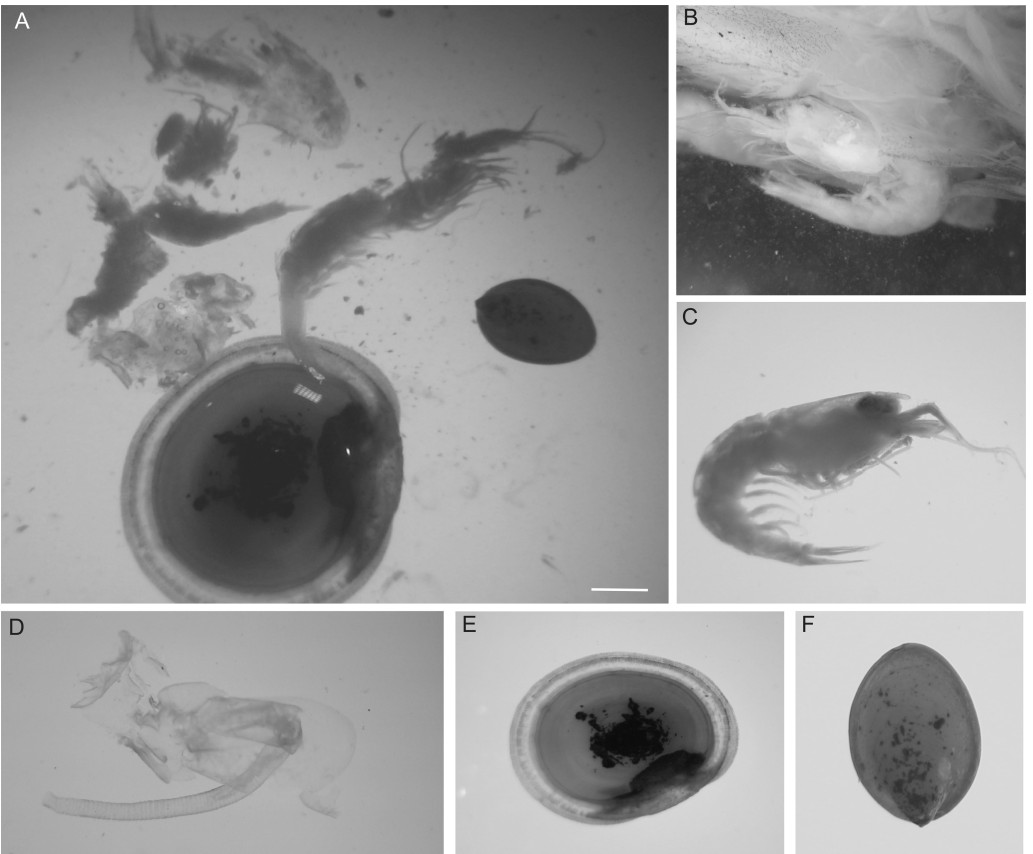

**Figure 1  Prey items in the gut of an individual eelpout, *Pyrolycus manusanus*, collected from South Su Manus Basin.** Stomach contents of an individual eelpout, *Pyrolycus manusanus*, collected from South Su Manus Basin. (A) Examples of prey items removed from the stomach showcasing (B) Alvinocaridids (shrimp) embedded in digestive tissue, (C) Alvinocaridids removed from stomach, (D) *Eochinoelasmus ohtai* (barnacle) penis, (E) *Shinkailepas tollmanni* (large limpet). (F) *Lepetodrilus schrolli* (small limpet). Scale bar: two mm.

eelpouts exceeding 16.5 cm in length preferred the large limpet, *Shinkailepas tollmanni* (Fig. 1E), while those under 16.5 cm preferred the barnacle, *Eochinoelasmus ohtai* (Fig. 1D), although there was no significant association between prey selection and eelpout body size for all prey (Table 2). The smallest eelpouts (<15.25 cm) preferred small limpets (*Lepetodrilus schrolli;* Fig. 1F). Alvinocaridid shrimp were present in 100% of eelpout stomach contents (Fig. 3) but was only observed in 52% of crab stomach contents (Fig. 2).

## DISCUSSION

Using stomach content analysis, deep sea organisms can be highly selective in their prey selection (*Ginger et al., 2001*; *Hudson et al., 2003*). Here, the eelpout, *Pyrolycus manusanus,* consumed alvinocaridid shrimp at a higher frequency relative to their abundance as they were the 4th most abundant organism in the South Su *Ifremeria* system. In contrast, limpets such as *Lepetodrilus schrolli* and *Shinkailepas tollmanni* were consumed at a lower frequency
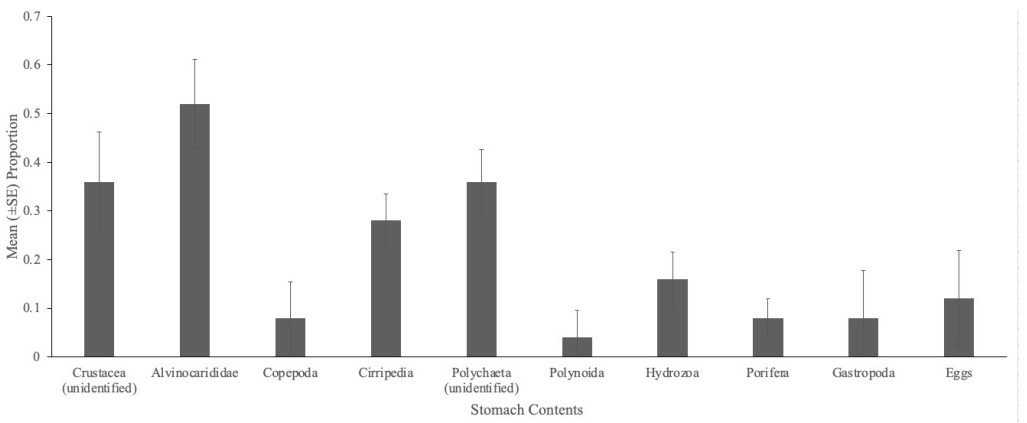

**Figure 2** **Gut contents of *Austinograea alayseae*.** Mean (±SE) proportion of brachyurans, *A. alayseae* (*n* = 25), containing various prey items during gut content analysis, collected at South Su, Manus Basin, Papua New Guinea.

**Table 2** **Assessment of prey selectivity in the eelpout *Pyrolycus manusanus* through the application of the Ivlev index (*E*).** Prey selectivity using the Ivlev index (*E*) for eelpout *Pyrolycus manusanus* (*n* = 7) at South Su, Ifermeria, Papua New Guinea. *E* values at 0 indicate random or no selectivity for a prey item, values above 0 indicate relatively higher prey selectivity, and values below 0 indicate lower prey selectivity. *R*-square values from regression analyses between the Ivlev Index (*E*) and eelpout body length (cm) for alvinocaridid shrimp, *Lepetodrilus schrolli* (small limpet), *Shinkailepas tollmanni* (large limpet) and *Eochionelasmus ohtai* (barnacle). n.s. indicates no significance (*P* < 0.05).

| Species | *E* | *R²* |
|---|---|---|
| Alvinocarididae | 0.94 | 0.18 n.s |
| *Shinkailepas tollmanni* | −0.52 | 0.09 n.s |
| *Lepetodrilus schrolli* | −0.80 | 0.15 n.s |
| *Eochinoelasmus ohtai* | −0.43 | 0.04 n.s |

despite being the most abundant invertebrate (11,360 and 858 individuals, respectively) within the South Su prey field. Our data show that the eelpout are selective predators. Preference for shrimp suggests that they are either of relatively higher nutrient quality, or easier to capture (*Sabelis, 1990*; *Underwood, Chapman & Crowe, 2004*). Given that all eelpouts had alvinocaridid shrimp in their diet, they are likely to be ecologically important to their feeding biology. This is further supported by *Machida & Hashimoto (2002)*, who collected eelpouts from the Manus Basin in Papua New Guinea and found alvinocaridid shrimp within their stomach contents. Similarly, a recently discovered species of *Pyrolycus* associated with a hydrothermal seep on the Pacific margin of Costa Rica was found to have a comparable diet, further underscoring the reliance of eelpouts on deep-sea shrimp (*Frable et al., 2023*).

*Austinograea alayseae* had a more diverse diet than their eelpout counterparts. They consumed 10 species of invertebrates compared to the four species consumed by eelpouts. Furthermore, crabs ingested polychaetes, which were never observed in the stomach contents of eelpouts. Such dietary preferences could be attributed to the morphology of

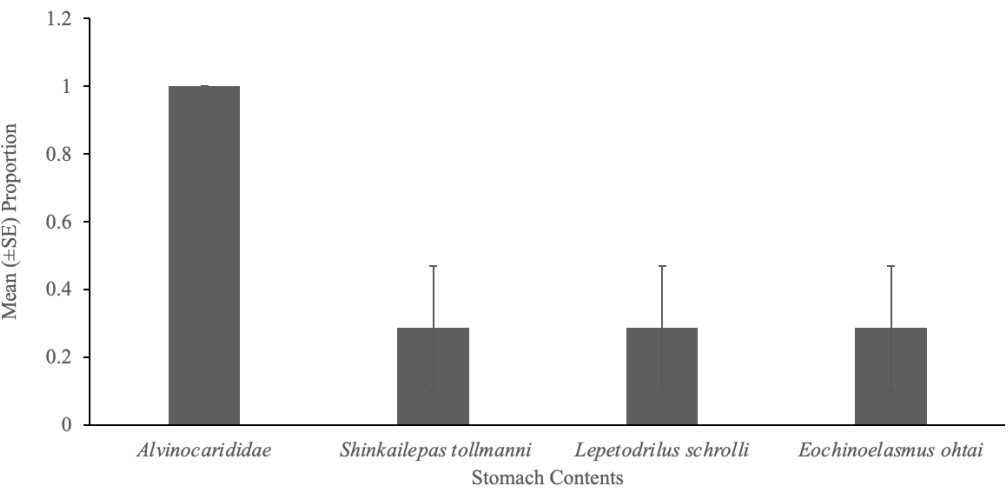

**Figure 3** **Gut contents of eelpout, *Pyrolycus manusanus*.** Mean (±SE) proportion of eelpouts, *Pyrolycus manusanus* ($n = 7$), containing various prey items during gut content analysis, collected at South Su, Manus Basin, Papua New Guinea.

their feeding structures (*Guinot, 1989*; *Machida & Hashimoto, 2002*). Using their chelipeds and mandibles, crabs are able to puncture and eviscerate the setae, cuticle and appendages of a polychaete (*Quammen, 1984*; *Petti, Nonato & Paiva, 1996*). Furthermore, none of their prey were observed intact, making it difficult to identify their prey to a low taxonomic level. Given their broad and less selective diet, our analysis supports that *Austinograea alayseae* is likely to be a generalist omnivore (*Van Audenhaege et al., 2019*) and are in a different feeding guild than eelpouts.

Our data show that eelpouts of all sizes had a strong preference for alvinocaridid shrimp with E values ranging from 0.9 to 1 but selected against all other prey taxa as a result of negative E values. However, eelpouts with a body length less than 16 cm preferred both species of limpets (*Shinkailepas tollmanni* and *Lepetodrilus schrolli*). Indeed, one individual had eight individual limpets (*L. schrolli)* within its stomach. Eelpouts possess a row of teeth that could aid them in prying off limpets from the substrate. Dietary observations from two eelpouts revealed that the only structure of the barnacle (*Eochinoelasmus ohtai*) present in their stomach was the annulated penis, with no other barnacle structures observed. This could be due to dissolvement of other barnacle structures before SCA was conducted. Given that the barnacle, *Eochinoelasmus ohtai*, was the most abundant barnacle observed in the study area, we infer that the annulated penis found in the stomach belong to this species. Indeed, one eelpout had three individual barnacle penises within its stomach. This suggests that either eelpouts are able to selectively ingest the barnacle penis while they are copulating, or that all other internal structures were digested well before the penis was digested.

We did not observe that eelpout size was significantly associated with prey selection, which may be attributed to our small sample size. However, we observed that larger individuals (more than 16 cm) tend to select shrimp and barnacles, while smaller individuals

(less than 16 cm) prefer to consume shrimp and *L. schrolli* limpets, which were the smallest prey items observed during stomach content analysis. Eelpouts may be selective predators as they consume co-occurring prey at different rates and sizes (*Ferry, 1997*; *Micheli et al., 2002*; *Gyldenskog, 2019*). This type of selection can be used to understand the importance of certain prey items to a predator's success, the structure of a food web (*Allesina, Alonso & Pascual, 2008*), as well as energy flow in a trophic cascade (*Ripple et al., 2016*).

The Ivlev index, while informative in understanding prey selectivity in certain studies, has faced challenges. For example, the Ivlev index assumes an even distribution of both predatory fish and their prey, which is not always the case (*O'Brien & Vinyard, 1974*). For instance, plankton can be aggregated within the water column, affecting prey selection for planktivorous fish (*O'Brien & Vinyard, 1974*). By applying the Ivlev index to stomach content analysis, obtaining unbiased samples that reflect prey abundances across habitats in which they are distributed can be difficult (*Strauss, 1979*). This can be particularly difficult in aquatic systems where traditional sampling methods for determining relative abundances and biomass of benthic invertebrates have been shown to be unreliable (*Strauss, 1979*). The index has been criticized as ineffective when relative abundances of prey vary across habitats (*O'Brien & Vinyard, 1974*; *Strauss, 1979*; *Collins, Kennedy & Van Dover, 2012*). Despite these challenges, the Ivlev index can still be informative, especially in cases where observing feeding behavior *in situ* is not feasible (*Rosca, Novac & Surugiu, 2010*). The variability of vent communities in space and time, along with the inherent bias of stomach content analyses toward hard-bodied organisms, may skew results. Future research could address these limitations by employing genetic barcoding to identify digested gut material more comprehensively (*Komura et al., 2018*).

## CONCLUSIONS

We hypothesized that eelpouts and crabs would be selective in their prey choice, despite co-occurring in the same deep sea hydrothermal vent community. We found that eelpouts had a strong preference for alvinocaridid shrimp, which supported by the high Ivlev index value. Crab prey selectivity was not quantified using the Ivlev index due the damaged condition of their stomach contents. Crabs were observed preying upon shrimp as well as polychaetes. By analyzing the stomach content of co-occurring eelpouts and crabs, instead of using inferences from isotopic data analysis, we are able to directly assess prey preference for these two common predators, providing insight on their trophic position and their feeding biology within the *Ifremeria* habitat. While this study was limited by sample size for both predators and quantification methods for crab prey selectivity, we show that eelpouts and crabs occupy different feeding guilds than originally described. Using a combination of stomach content analysis, isotopic data, and other metrics (*e.g.*, video evidence of predation *etc.*), future research can assess the role of such predators in the food webs of deep-sea hydrothermal vent communities.

## ACKNOWLEDGEMENTS

All samples were analyzed and collected on behalf of the people of Papua New Guinea. Samples were provided courtesy *via* Dr. Cindy Van Dover's archives. Logistical support for this project was provided by Eckerd College.

### Funding

This work was supported by Duke University Marine Lab's Marine Science and Conservation Scholarship. The funders had no role in study design, data collection and analysis, decision to publish, or preparation of the manuscript.

### Grant Disclosures

The following grant information was disclosed by the authors:
Duke University Marine Lab's Marine Science and Conservation Scholarship.

### Competing Interests

The authors declare there are no competing interests.

### Author Contributions

- Deidric B. Davis conceived and designed the experiments, performed the experiments, analyzed the data, prepared figures and/or tables, authored or reviewed drafts of the article, and approved the final draft.
- Nancy Smith analyzed the data, prepared figures and/or tables, authored or reviewed drafts of the article, and approved the final draft.

### Animal Ethics

The following information was supplied relating to ethical approvals (i.e., approving body and any reference numbers):

We consulted with the Duke University IACUC committee on 9 July 2024 and learned that "The IACUC does not require approval for dead, vertebrate, animal activities unless the activity involves (1) killing animals for the purpose of obtaining or using their tissues or other materials, or (2) project-specific antemortem manipulation of animals prior to killing them". Reference OLAW FAQ A.3. While the fish were collected at depth, their collection was unintentional and not noted by the observers aboard the surface ship until the samples were brought on board and the dead fish that did not survive the 1,400 m ascent were discovered.

### Data Availability

The raw data is available in the Supplementary File.

### Supplemental Information

Supplemental information for this article can be found online at http://dx.doi.org/10.7717/peerj.19476#supplemental-information.

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
