# Peer review of "Diet and prey selectivity in co-occurring eelpout fish and bythograeid crabs in a deep-sea hydrothermal vent community"

_PeerJ, doi:10.7717/peerj.19476_

## Round 0.1 · original submission · Major Revisions

Dear Authors,

Your article has been evaluated by several referees and a ‘Major revision’ decision has been made. Please review the referee's suggestions and corrections carefully.

You can see the reviewers' comments on your article below.

I wish you success in your work.

Reviewer 1 ·

Basic reporting

I have presented all my views regarding the article in part 4.

Experimental design

I have presented all my views regarding the article in part 4.

Validity of the findings

I have presented all my views regarding the article in part 4.

Additional comments

Overview of the article entitled “Prey selectivity and gut content analysis in cooccurring brachyurans and eelpout fish in a deep sea
hydrothermal vent community”.

The study focuses on the feeding ecology of two predator groups (eelpouts and brachyurans) in deep-sea hydrothermal vent communities. By analyzing their diets and prey selectivity, we aim to understand their trophic interactions and roles in the ecosystem. The main findings show that eelpouts strongly prefer Chorocaris shrimp, while brachyurans have a more diverse diet, suggesting scavenging or omnivorous feeding behavior. The study highlights the importance of these predator-prey dynamics in understanding the ecological structure and functioning of deep-sea environments, especially in the context of hydrothermal vent ecosystems. The study has been prepared meticulously in general and will be acceptable for publication if the following single advice is taken into consideration.

- The introduction should include basic bioecological information (diet composition, spawning times, maximum length, etc.) about Pyrolycus manusanus and Austinograea alayseae species, in addition to their morphological characteristics.

Reviewer 2 ·

Basic reporting

-Language used throughout this manuscript.is clear.
-Background is enough to overview the topic.
The manuscript structure is in good flow.
-Raw data were shared.
The references used in the manuscript are sufficient and recent.

Experimental design

The experimental design was well designed and methodologies were used that were appropriate for the purpose of the study. However, I would expect the stomach content data to be evaluated with different indices. For example, the Index of Relative Importance (IRI) of Pinkas et al. (1971). Also, the authors should have determined the fractional trophic level (TROPH) values of species were estimated using TrophLab and the Pauly et al. (2000) equation.
Additionally, I wish the stomach contents were analyzed by molecular methods.

I would like to point out that what I have stated above is not a deficiency in this study.
On the contrary, I think that since researchers have obtained very valuable data, therefore, these data should be evaluated in many ways.
However, I am aware of the constraints of time and funding.

Validity of the findings

The findings in the study are extremely valuable and will be a unique resource for future research. Extreme ecosystems such as hydrothermal vents have been overlooked for many reasons and are not studied much.

Additional comments

If possible, please use a scientific name in Fig 2. (not sponge use Porifera or not shrimp use Malacostraca, e.g..)
Check the spelling of sp. in the text.

Reviewer 3 ·

Basic reporting

The English in the manuscript is clear, but some sections seem repetitive and somewhat lengthy. I would recommend restructuring particularly in the introduction to improve clarity and conciseness in conveying the purpose of the study. Additionally, the limitation of stable isotope analysis and the need for the gut content analysis should be explained more clearly.
I recommend adding more references to the related parts on the ecosystem structure and food sources of vent organisms. For instance, Suh et al. (2022) in Marine Biology revealed the food web structure of dominant hydrothermal vent animals in the Western Pacific, including analysis results for A. alayseae, the same crab species. Additionally, as a minor point, the reference at line 164, Kise et al. (2013), seems mismatched, as the article focuses on genetic connectivity, which is unrelated to the scavenging behavior of crab species.

Experimental design

As I mentioned earlier, the need for gut content analysis is not clearly explained. It would be helpful to elaborate on why this analysis is necessary and how it complements the stable isotope analysis.
I have some concerns that the current experimental design may not be fully sufficient to accurately determine the prey preferences of the two species. Factors such as the rate (e.g., speed) of digestion and the varying relative abundance of prey across different habitats could potentially influence the results. I kindly suggest that these aspects be considered more carefully. Additionally, the statistical significance of prey preferences should be assessed.
Regarding prey identification, I assume the authors identified prey taxa based on morphology by comparing the species list from the previous study by Collins, Kennedy, & Van Dover (2012). However, more detailed explanations should be provided in the methods section, such as how the authors distinguished between the two shrimp genera, Chorocaris and Alvinocaris (as both were reported in the same habitat per Collins, Kennedy, & Van Dover, 2012), how they identified the barnacle species penis (Fig. 1B), and the criteria used to separate unidentified prey. Additionally, higher resolution and larger images should be included. As a minor point, a scale bar should be added to Fig. 1 for clarity.

Validity of the findings

This manuscript is interesting; however, there are a few important concerns, such as the statistical significance of the data and the alignment of the experimental design with the hypothesis, that should be carefully addressed before publication. Additionally, one of the major results regarding the correlation between predator size and prey preference is not statistically supported, and therefore should not be presented as a key finding.

Reviewer 4 ·

Basic reporting

The authors describe diet information on two poorly-known deep-sea organisms, an eelpout and a brachyuran crab (neither of which seems to have a non-scientific common name). While the scope of the conclusions is limited by the sample sizes, the analyses were conducted using standard methods and provide novel information. The availability of raw observational data also allowed an unusual but helpful additional analysis via Ivlev's Index regarding prey selectivity.

However, the authors should use care within the manuscript to avoid conflating their two individual species with additional species from these respective taxa. In particular, Infraorder Brachyura "true crabs" encompasses approximately 7000 species, most of which bear little ecological similarity with Austinograea alayseae; using "brachyuran" as a shorthand for this deep-sea species seems misleading, even if unintentional. If the authors wish an alternative for the scientific names, "eelpout" and "crab" would suffice (and as the authors themselves do in Table 2).

Otherwise, the manuscript was clearly written with sufficient literature and background. Some minor suggestions have been made regarding the text in the "Additional comments" section below.

Experimental design

This was essentially a descriptive study, so there was little explicit experimental design. However, the authors need more information regarding their samples, such as mean size (+/- SD). Many fishes in particular exhibit ontogenetic diet changes, so size is a very relevant metric, especially since the authors themselves present a size-based analysis in L133-137 (and discussed in L170-172). The authors are also encouraged to provide context to these sizes (i.e., how representative they are to the overall populations at this location).

Validity of the findings

Given the difficulty in obtaining deep-sea specimens, the low number of samples (for both species) is not concerning, although it would be interesting to compare these stomach content findings against either other eelpout species or deep-sea (vent-associated) fishes in general.

Additional comments

L47: the sentence should read "Eelpouts (Family Zoarcidae) are predatory fishes..."
L51: replace "fish" with "eelpout"
L52-55: unclear why this information is relevant to the topic of the paper
L58-59: similarly, the sentence should read "... (Infraorder Brachyura, Family Bythograea) is a blind brachyuran crab that inhabits hydrothermal vents."
L82-84: this sentence should be in Results (and implications possibly reviewed in Discussion)
L104-105: surprising that these observations weren't at least reported and discussed qualitatively, since plastics in deep-sea environments are still poorly known
L125 (and elsewhere): "sp." should not be italicized
L137: for consistency, "Chorocaris (shrimp)" should simply be "Chorocaris sp.", as the authors have clearly noted this taxa as shrimp earlier
L149-153: these sentences should be combined or reworded
L156-160: this section on polychaetes highlights the reason that relative sizes are important; a 16.5 cm fish should be able to ingest a polychaete, unless these deep-sea species are heavily armored or unusually large
L163: why discuss sulfides if they're not presented (see L104-105)?
L174-179: it seems strange that this would be the only structure remaining, especially if eelpouts have a strong enough jaw structure to remove limpets (L173)
L185-187: this sentence should be reworded for clarity
L192-196: these sentences should be combined

Reviewer 5 ·

Basic reporting

Gut content analysis is one of the useful methods to understand in all environments and ecosystems. But, overall, the manuscript is problematic, rambling, incoherent and lacking polish. In addition to, it would greatly benefit from English proofreading the manuscript. Please check the detailed comments in the attached file.

Experimental design

a. In the manuscript, the authors repeatedly discussed the relationship between body size and prey in eelpouts. However, I believe they did not achieve their objectives in this study
b. The terms for the two organs, gut and stomach, are confused in all text, figures, Tables.
c. The authors said that the quantification of gut contents in crabs but they presented Fig. 2. How do you make Fig 2?
d. In a food web at a single site, cohibiting species make efforts to avoid prey competition, making the identification of prey at the species level crucial. Therefore, you should verify this using CO1 DNA barcodes through Sanger sequencing or metabarcoding with NGS. Were all specimens fixed in formalin? If so, this may have hindered the study from achieving its objectives.

Validity of the findings

n/s

Additional comments

The introduction section seems to be write based on incomplete or inaccurate information. Please ensure the proper definition of keywords and rewrite this section clearly and concisely. Specifically, species names need to be revised using valid names, refer to WoRMS for accuracy.

Annotated reviews are not available for download in order to protect the identity of reviewers who chose to remain anonymous.

Reviewer 6 ·

Basic reporting

The study is a well-reported examination of diet preference and gut content of two high-trophic-level scavengers at Manus Basin hydrothermal vents. The methods for examining gut contents and the discussed scope and relevance of the findings is appropriate. The study was taken a step further by comparing prey frequency in gut contents to faunal relative abundance in the vent community to infer prey selectivity. The writing was clear and unambiguous. Detailed comments and grammatical edits are provided in the attached file pdf.

The literature references were appropriate; however, some key relevant studies were omitted. For example, the following studies should be cited and discussed, as they examine prey selectivity of other hydrothermal vent fish:

Sancho, G., Fisher, C.R., Mills, S., Micheli, F., Johnson, G.A., Lenihan, H.S., Peterson, C.H. and Mullineaux, L.S., 2005. Selective predation by the zoarcid fish Thermarces cerberus at hydrothermal vents. Deep Sea Research Part I: Oceanographic Research Papers, 52(5), pp.837-844.

Micheli, F., Peterson, C.H., Mullineaux, L.S., Fisher, C.R., Mills, S.W., Sancho, G., Johnson, G.A. and Lenihan, H.S., 2002. Predation structures communities at deep‐sea hydrothermal vents. Ecological monographs, 72(3), pp.365-382.

Further studies dealing with vent predation or food webs in general could be included:

Voight, J.R., 2000. A review of predators and predation at deep-sea hydrothermal vents. Cahiers de biologie marine, 41(2), pp.155-166.

Bergquist, D.C., Eckner, J.T., Urcuyo, I.A., Cordes, E.E., Hourdez, S., Macko, S.A. and Fisher, C.R., 2007. Using stable isotopes and quantitative community characteristics to determine a local hydrothermal vent food web. Marine Ecology Progress Series, 330, pp.49-65.

Buckman, K.L., 2009. Biotic and abiotic interactions of deep-sea hydrothermal vent-endemic fish on the East Pacific Rise (Doctoral dissertation, Massachusetts Institute of Technology).

There are now three described species in Pyrolycus:

Frable, B.W., Seid, C.A., Bronson, A.W. and Møller, P.D.R., 2023. A new deep-sea eelpout of the genus Pyrolycus (Teleostei: Zoarcidae) associated with a hydrothermal seep on the Pacific margin of Costa Rica.

In several places, I caught spelling errors of scientific names, or names that are no longer accepted. Please refer to the World Register of Marine Species to check the validity of all Latin names in the manuscript. https://www.marinespecies.org/

The article structure was professional and the flow of reasoning clear. The article was self-contained with relevant results to hypotheses. I think you could structure a stronger hypothesis by adding a paragraph to the introduction and discussion to frame the study around foundational concepts like the competitive exclusion principle and resource partitioning, which aim to explain how species of similar ecological niches might coexist. For example:

Gause, G.F., 2019. The struggle for existence: a classic of mathematical biology and ecology. Courier Dover Publications.

Schoener, T.W., 1974. Resource Partitioning in Ecological Communities: Research on how similar species divide resources helps reveal the natural regulation of species diversity. Science, 185(4145), pp.27-39.

Experimental design

This study presents primary original research and is within the Aims and Scope of PeerJ by being scientifically and methodologically sound and providing a thoughtful and encompassing treatment of the subject material.

The research question is well defined and fills a knowledge gap because any information on the dietary preferences of deep-sea hydrothermal vent species is illuminating and important for further work on their habitat requirements to inform conservation. However, I recommend that the authors bring more ecological theory on feeding niches into the introduction and discussion. This will help make a more direct connection to how having distinct feeding preferences might influence community structure and allow for species coexistence. For example, review the below references for how prey selectivity might influence community structure.

Micheli, F., Peterson, C.H., Mullineaux, L.S., Fisher, C.R., Mills, S.W., Sancho, G., Johnson, G.A. and Lenihan, H.S., 2002. Predation structures communities at deep‐sea hydrothermal vents. Ecological monographs, 72(3), pp.365-382.

Sancho, G., Fisher, C.R., Mills, S., Micheli, F., Johnson, G.A., Lenihan, H.S., Peterson, C.H. and Mullineaux, L.S., 2005. Selective predation by the zoarcid fish Thermarces cerberus at hydrothermal vents. Deep Sea Research Part I: Oceanographic Research Papers, 52(5), pp.837-844.

The authors seem to have conducted all the methodological processes accurately and provide sufficient detail to replicate. The authors were working with fish caught as bycatch, so explicit replication was not planned. However, they have enough fish to show their results are not random chance. The community matrix used for prey selectivity comparison is from a well-designed study with good replication.

Validity of the findings

All data is presented. I will not ask that you change the supplementary data format, but for future studies data in “long” format is easier for other users to work with. For example, having one column being “fish individual” another be “abundance” another be “relative abundance” etc, so that there aren’t many separate tables for each metric. In the supplementary data table, I was also confused what “length” and the table next to “length” are. Please make this clearer so future users can interpret the data appropriately.

Main concerns I have with the conclusions based on the methodology are as follows:

1) The inability to identify digested material and the bias towards species with hard shells. Please add more discussion on what aspects of the diet might be missed if soft-bodied organisms are already digested at the time of dissection. Would bias results more towards gastropods and crustaceans, and away from polychaetes. For future direction, DNA barcoding could be used to identify digested gut contents, but I understand if the authors do not have these resources for the present study.

2) The spatial and temporal variability in the vent community. All gut content analyses are just a snapshot of food that the animal ate in the last several days. The study cited for the community matrix explicitly discusses and accounts for extreme variability (“contagiousness”) in vent community abundance, which might influence selectivity depending where and when a fish is feeding. This is the reality of gut content analyses; I am not saying the author’s methods were flawed and I still think any deep-sea results are better than none. I would encourage the authors to highlight more in the discussion that feeding selectivity results might be less robust in systems like vents with high spatial and temporal variability. Also, please explain in more detail in the Methods whether/how the raw faunal abundance table was subset or transformed for the Ivlev analysis. For example, was abundance calculated by adding together all sites for the Ifremeria habitat, or taken from the single site closest to where the fish were collected?

Additional comments

Overall, I think the study is very good and well-written, and should be published following a major revision.

Annotated reviews are not available for download in order to protect the identity of reviewers who chose to remain anonymous.

---

## Round 0.2 · Minor Revisions

Dear Authors,

Your article has been evaluated by several referees and a 'Minor revision' decision has been made. Please carefully review the reviewers' suggestions and make the necessary corrections.

Note: In particular, the referees' question about the relationship between predator size and prey preference should be answered in more detail and the method of taxon identification of prey should be explained more clearly.

Best Regards
Servet

Reviewer 1 ·

Basic reporting

-

Experimental design

-

Validity of the findings

-

Additional comments

I've checked the revised M&S and accepted it for publication in its final version.

Reviewer 3 ·

Basic reporting

The authors have made an effort to address my review, and I appreciate their revisions. However, the revised manuscript does not fully address all of my concerns.

Experimental design

This study provides valuable insights into the ecological functions of the deep-sea ecosystem, and I acknowledge the challenges of studying vent organisms, particularly in terms of sampling. However, the methods need further strengthening and refinement to ensure the study meets the standards for publication, especially in a high-ranking journal.

Validity of the findings

I still do not fully understand the response to my question about the association between predator size and prey preference, as the result is not supported by a statistical analysis.

Additional comments

As I mentioned in my first review, the method for taxa identification of prey should be explained more clearly. Although the authors stated that additional details were included in the Methods section, I was unable to find them.

Additionally, I have a few minor comments below:

Please check the table numbers in lines 212 and 214.
Please specify the scale bar size in Figure 1.
Please ensure consistency in taxa names. For example, the shrimp is referred to as Alvinocarididae in the main manuscript but as Chorocaris sp. in Figure 1. Similarly, "Olgasolaris tollmani" should be modified to match the name used in the main manuscript.

---

## Round 0.3 · accepted · Accept

Dear Dr. Davis and Dr. Smith,
Thank you for editing your publication in accordance with the reviewer's suggestions. Your manuscript has been "accepted" for publication. I wish you success in your future studies.